# SARS-CoV-2 Infection and Cardioncology: From Cardiometabolic Risk Factors to Outcomes in Cancer Patients

**DOI:** 10.3390/cancers12113316

**Published:** 2020-11-10

**Authors:** Vincenzo Quagliariello, Annamaria Bonelli, Antonietta Caronna, Gabriele Conforti, Martina Iovine, Andreina Carbone, Massimiliano Berretta, Gerardo Botti, Nicola Maurea

**Affiliations:** 1Division of Cardiology, Istituto Nazionale Tumori-IRCCS-Fondazione G. Pascale, 80131 Napoli, Italy; a.bonelli@istitutotumori.na.it (A.B.); a.caronna@istitutotumori.na.it (A.C.); g.conforti@istitutotumori.na.it (G.C.); martina.iovine@istitutotumori.na.it (M.I.); andreina.carbone@istitutotumori.na.it (A.C.); 2Department of Medical Oncology-Centro di Riferimento Oncologico di Aviano (CRO), IRCCS, 33081 Aviano, Italy; mberretta@cro.it; 3Scientific Direction, Istituto Nazionale Tumori-IRCCS-Fondazione G. Pascale, 80131 Napoli, Italy; g.botti@istitutotumori.na.it

**Keywords:** COVID-19, inflammation, cytokines, myocardial injury, cardiovascular diseases, cardioncology

## Abstract

**Simple Summary:**

The coronavirus disease-2019 (COVID-19) pandemic has significantly changed the management and treatment of some diseases, including cancer, in order to reduce the risk of viral contamination in particularly vulnerable patients. SARS-CoV-2 infection leads to secondary hemophagocytic lymphohistiocytosis (sHLH), which is a multiorgan hyperinflammatory condition based on the hyperactivation of cytotoxic T lymphocytes, macrophages, and natural killer cells, leading to multiorgan failure (including myocarditis, venous thromboembolism, and acute respiratory distress syndrome) and consequently to death. We highlight the major cardiovascular and coagulative complications of COVID-19 with particular reference to cancer patients by analyzing the retrospective clinical studies currently available. Discussions on the harmful or beneficial effects of anticancer therapies as well as on the type of tumor during SARS-CoV-2 infection have been made, with the addition of practical recommendations for the risk reduction of coagulation dysfunctions, myocarditis, venous thromboembolism, and mortality in cancer patients.

**Abstract:**

The coronavirus disease-2019 (COVID-19) is a highly transmissible viral illness caused by SARS-CoV-2, which has been defined by the World Health Organization as a pandemic, considering its remarkable transmission speed worldwide. SARS-CoV-2 interacts with angiotensin-converting enzyme 2 and TMPRSS2, which is a serine protease both expressed in lungs, the gastro-intestinal tract, and cardiac myocytes. Patients with COVID-19 experienced adverse cardiac events (hypertension, venous thromboembolism, arrhythmia, myocardial injury, fulminant myocarditis), and patients with previous cardiovascular disease have a higher risk of death. Cancer patients are extremely vulnerable with a high risk of viral infection and more negative prognosis than healthy people, and the magnitude of effects depends on the type of cancer, recent chemotherapy, radiotherapy, or surgery and other concomitant comorbidities (diabetes, cardiovascular diseases, metabolic syndrome). Patients with active cancer or those treated with cardiotoxic therapies may have heart damages exacerbated by SARS-CoV-2 infection than non-cancer patients. We highlight the cardiovascular side effects of COVID-19 focusing on the main outcomes in cancer patients in updated perspective and retrospective studies. We focus on the main cardio-metabolic risk factors in non-cancer and cancer patients and provide recommendations aimed to reduce cardiovascular events, morbidity, and mortality.

## 1. Introduction

In December 2019, a previously unknown coronavirus called Severe Acute Respiratory Syndrome coronavirus 2 (SARS-CoV-2) [1] caused severe acute respiratory syndrome in Whuan, the capital of Hubei, in China. In March 2020, the World Health Organization (WHO) defined the SARS-CoV-2 infection as a pandemic, due to its high rate of virulence and propagation in the world, which could potentially affect millions of people [2]. The full spectrum of the disease, termed coronavirus disease-2019 (COVID-19), ranges from mild, self-limiting respiratory tract illness to severe progressive pneumonia, multiorgan failure, and death. To 28 October 2020, the virus spread to over 150 countries and affected more than 44 million individuals, causing over one million deaths. Coronaviruses are a large family of zoonotic, enveloped, and non-segmented RNA-based viruses, which cause illness ranging from common cold to severe respiratory diseases [3,4]. Other similar coronavirus-related infections in humans are Severe Acute Respiratory Syndrome (SARS-Cov) [5] and Middle East respiratory syndrome-related coronavirus (MERS-Cov) [6]. The mortality rate for 10,000 total cases were 10 and 37% for SARS-Cov and MERS-Cov, respectively [6,7]. Notably, patients affected by SARS-Cov and MERS-Cov had cardiovascular complications such as myocardial infarction and fulminant myocarditis [8]. SARS-CoV-2 infection causes multiorgan damages involving lungs and the cardiovascular system [9] and patients with chronic diseases such as cancer, hypertension, diabetes, cardiovascular diseases, liver steatosis, asthmatic bronchitis, and chronic inflammatory diseases are particularly vulnerable [10,11,12]. Cardiovascular diseases such as ischemic heart disease, stroke, heart failure, and peripheral arterial disease are the leading cause of global mortality [13]. In 2017, cardiovascular diseases caused an estimated 17.8 million deaths worldwide [13]. Improvements in cancer survival in the past few decades have resulted in a large and growing population of long-term cancer survivors; about half of patients diagnosed with cancer in high-income settings are now expected to survive for 10 years or longer [14]. The increase in survival is associated with a high risk of cardiovascular diseases related to anticancer therapies. The cardiotoxic effects of chemotherapy, radiotherapy, and immune checkpoint inhibitors expose cancer survivors to high cardiac and vascular vulnerability [14]. Very recent data indicate that patients with underlying cardiovascular diseases or cancer have a high risk of SARS-CoV-2 infection and a poorer prognosis than healthy people [9]. We highlight the cardiovascular side effects of SARS-CoV-2 infection, its pathogenesis of multiorgan damages, and the main outcomes in cancer patients currently available (Figure 1). Moreover, we discuss the cardio-metabolic risk factors in COVID-19, providing recommendations to reduce the incidence of viral infection and mortality in subjects who are particularly vulnerable such as cancer patients. The literature search was performed using the PubMed database and the Cochrane library with the following research items: “novel coronavirus”, “2019-nCoV”; “COVID-19”; “SARS-CoV-2”; all of them followed with “AND cardiovascular diseases or heart or myocardial diseases or cardiac injury or cardiovascular system”.

## 2. Cardiovascular Outcomes in Non-Cancer Patients with COVID-19

Reports suggest that SARS-CoV-2 and MERS-CoV could have the same effects in humans with similar multiorgan damages leading to cardiovascular adverse events (more frequently heart failure and myocarditis) and severe pneumonia [9]. In Figure 2, we summarized the risk and prognostic factors in COVID-19 and the most frequent seen cardiovascular adverse events. In brief, risk factors of SARS-CoV-2 infection are active cancer, previous cardiotoxic anticancer therapies, metabolic syndrome, prior cardiovascular diseases, type 2 diabetes mellitus, hypertension, coagulopathy, shock, smoke, low or no physical activity, gene polymorphism of ectoenzyme angiotensin-converting enzyme 2 (ACE2). Prognostic factors of SARS-CoV-2 infection are d-Dimer greater than 1 μg/mL, older age, high Sequential Organ Failure Assessment score (also called sepsis-related organ failure assessment score, involving the health status of respiratory, cardiovascular, hepatic, coagulation, renal, and neurological systems), high levels of high-sensitivity cardiac troponin I, lactate dehydrogenase, and interleukin-6 as well as low levels of lymphocytes (lymphopenia) [15]. Patients with COVID-19 have multiple cardiovascular adverse outcomes involving fulminant myocarditis, venous thromboembolism, hypertension, arrhythmia, heart failure, cardiogenic shock, and acute coronary syndrome [1,9]. In China, 12% of patients without underlying cardiovascular diseases experienced heart damages with increased levels of cardiac troponin levels and d-dimer or cardiac arrest during hospitalization for COVID-19 [1]; it was the first evidence of extra-pulmonary effects of SARS-CoV-2 in humans. Moreover, high levels of pro-inflammatory markers were seen in these patients, confirming the inflammatory nature of the coronavirus-associated clinical outcomes [1]. Another evidence of cardiovascular effects of SARS-CoV-2 infection was described by Chen et al., where 150 patients with COVID-19 admitted to Intensive Care Units (ICU) correlated with multiple clinical factors i.e., older age, high levels of hypersensitive C-reactive protein (hs-CRP), serum creatinine, NT-proBNP, and cTnI, as well as a history of hypertension and coronary heart diseases [10]. Chen et al. clearly demonstrated a correlation between SARS-CoV-2 and heart, and they concluded that two independent determinants of poor prognosis in COVID-19 patients were the past medical history of coronary heart disease and increased levels of cardiac troponin-I [16]. Another study on 138 patients with COVID-19 demonstrated that 16.7% of total patients and 44.4% of ICU patients developed arrhythmia, respectively, and 7.2% of total patients experienced acute cardiac injury in addition to other COVID-19 related complications. Moreover, 58% of patients had hypertension and 25% had heart diseases [17]. Indeed, ICU patients had higher biomarkers of myocardial injury compared to non-ICU patients (increases in 30% of median creatine kinase (CK)- myocardial band (MB) and 60% of hypersensitive circulating troponin levels (*p* < 0.001; *p* = 0.004, respectively) [17]. In another study [1] on 41 patients with COVID-19, Huang et al. demonstrated that 12.2% of patients developed acute cardiac injury with increased levels of high-sensitivity cardiac troponin I (hs-cTnI) and 80% of them were admitted to the ICU and had hypertension with a mean systolic blood pressure of 145 mmHg (19% more than non-ICU COVID-19 patients, *p* < 0.001) [1]. 

## 3. Outcomes in Cancer Patients

Cancer survivors and patients with active cancer have a considerably increased risk for premature cardiovascular and infection diseases, mainly due to cardiotoxic cancer treatments and immunosuppression state [18]. Patients with active cancer and cancer survivors are extremely vulnerable to viral infection, and frequently, they have a poor prognosis of infection diseases [19]. Considering that SARS-CoV-2 infection exerts cardiovascular side effects and that cancer survivors may have been treated with cardiotoxic anticancer treatments, it is deductible that patients with cancer may have a poor prognosis of COVID-19 than non-cancer patients. According to the clinical guide for the management of non-coronavirus patients of the National Health Service England [20] (Figure 3), the main risk factors of SARS-CoV-2 infection in cancer patients are the active treatments with chemotherapy and radiotherapy for the treatment of lung cancer; leukemia, lymphoma, or myeloma (under any type of treatment); cancer patients treated with immune checkpoint inhibitors; cancer patients treated with protein kinase inhibitors or poly ADP ribose polymerase (PARP) inhibitors; patients treated with transplants of bone marrow or stem cells (in the last 6 months), or under treatment with immunosuppression drugs. Moreover, the same document of The National Health Service England in collaboration with The European Society of Medical Oncology suggested that oncologists should give higher priority to treat cancer patients clinically unstable and consider to change intravenous treatments to subcutaneous or oral administrations (possibly with longer intervals between cancer treatments (immune checkpoint inhibitors, or ICIs) with the use of granulocyte colony-stimulating factor (GCSF) as primary prophylaxis in these patients [20].

Retrospective studies on cancer patients with COVID-19 are summarized in Table 1. A first study of Liang W et al. [21] on 1580 Chinese patients with COVID-19 reports that 1% of them had a history of cancer (95% CI 0.61–1.65). Cancer patients have a higher probability of severe events than non-cancer patients (hazard ratio 3.56, 95% CI 1.65–7.69). Cancer patients with recent chemotherapy or surgery have a higher risk of clinically severe events than the other patients (odds ratio 5.34, 95% CI 1.80–16.18; *p* = 0.0026), indicating that recent pharmacological or surgical interventions may adversely affect the prognosis of COVID-19 in these patients. Jing Yu et al. [22] described in 1524 cases that patients with cancer had a higher risk of SARS-CoV-2 infection (OR, 2.31; 95% CI, 1.89–3.02) compared with other people, and Guan et al. [23] reported that cancer patients had a hazard ratio of 3.50 (95% CI, 1.60–7.64) for admission to ICU, invasive ventilation, or death, compared to non-cancer patients. A recent publication of the Istituto Superiore di Sanità (Italy) stated that among 909 patients who died, more than 50% had three or more comorbidities such as ischemic heart disease (27.4%), atrial fibrillation (23%), heart failure (16.4%), stroke (12%), hypertension (73.5%), diabetes mellitus (31.5%), and active cancer in the past 5 years (16.5%) [24]. Trapani et al. [25] described that cancer patients with COVID-19 had a smoking history, being either former smokers or current smokers. Comorbidities were arterial hypertension, diabetes mellitus type II, and chronic kidney diseases. Italian study [26] demonstrated that male cancer patients had an increased risk of SARS-CoV-2 infections than patients without cancer (Relative Risk 0.2% and 0.3%, respectively). Interestingly, prostate cancer patients receiving androgen-deprivation therapies had a significantly lower risk of SARS-CoV-2 infection compared to patients who did not receive androgen-deprivation therapies (OR 4.05; 95% CI 1.55–10.59). In sum, male cancer patients have an increased risk of SARS-CoV-2 infection and develop more severe forms of COVID-19, which is in line with other studies. This study highlights the role of androgens in risk of SARS-CoV-2 infection; prostate or breast cancer patients with high circulating androgens levels may have an increased risk of viral infection and a poor prognosis. TMPRSS2, a member of the family of Type II transmembrane serine proteases, is expressed on lung, prostate, and breast cancer cells, and its expression is regulated by androgen receptors [27]; the authors discussed potential TMPRSS2 targeting through pharmacological approaches. A more recent publication of de Rojas et al. [28] stated that the prevalence of COVID-19 infection among children with cancer in Madrid was 1.3%, while that in the general pediatric population is 0.8%.

Another study of Wu et al. [29] on 72,314 Chinese patients demonstrated that patients with pre-existing comorbidities had a higher risk of death: 10.5% for cardiovascular disease, 7.3% for diabetes, 6.3% for chronic respiratory disease, 6.0% for hypertension, and 5.6% for cancer. An Italian report made by Onder et al. [30] on 1625 patients with COVID-19 showed that in a subsample of 355 patients who died, 30% of them had ischemic heart disease, 35.5% had diabetes, 20.3% had active cancer, 24.5% had atrial fibrillation, 6.8% had dementia, and 9.6% had a history of stroke. A retrospective cohort study on 28 COVID-19-infected cancer patients [31] (derived from a sample of 1276 subjects with COVID-19) support the vulnerability of these patients for SARS-CoV-2 infection with a mortality rate of 28.6% compared to 2–3% of patients without cancer in China. The authors analyzed that 60.7% of patients were male, the median age was 65 years, 25% had lung cancer, 14.3% had esophagus cancer, 10.7% had breast cancer, 7.1% had laryngocarcinoma, liver, and prostate cancer. Of note, 75 and 89% of patients recently had surgery or chemotherapy/radiotherapy, respectively; 21.4% of patients received target/immunotherapy, and 10.7% received chemotherapy in the past 2 weeks. Notably, 14.3% of cancer patients had diabetes or chronic cardiovascular and cerebrovascular disease (including hypertension and coronary heart diseases). Moreover, an independent predictor of death was the treatment with chemotherapy, immunotherapy, or radiation within 2 weeks before SARS-CoV-2 infection (hazard ratio > 4). A case report of Bonomi et al. [32] described an event in a patient with metastatic non-small-cell lung cancer in treatment from 6 years with anti Programmed Death type 1 (PD-1) antibody nivolumab. After COVID-19 diagnosis and hospitalization, the patient presented a rapid evolution of respiratory failure. The authors discuss the need for a multidisciplinary approach for the management of COVID-19-infected cancer patients, opening a possible discussion on the role of immune checkpoint inhibitors in COVID-19 prognosis. He et al. [33] report data of 128 patients with COVID-19; there was no significant difference in the proportion of patients with hematological cancers vs. healthy subjects (10% vs. 7%, *p* = 0.322), but a significant difference was seen in fatality rate (62% vs. 0%, *p* = 0.002). Among cancer patients, 39% had acute myeloid leukemia, 20% had acute lymphoblastic leukemia, and 59% had received chemotherapy in the last 7–19 days.

Miyashita [34] et al. showed that cancer patients had a higher relative risk of intubation for SARS-CoV-2 infection of 1.89 (95% CI 1.37–2.61) than non-cancer patients, and that younger cancer patients had a significantly higher rate of mortality than the older (RR 5.01, 95% CI 1.55–16.2).

Another study of Dai et al. [35] on 641 Chinese patients stated that patients with cancer had a higher mortality rate than non-cancer patients (odds ratio 2.34, 95% CI 1.15–4.77, *p* = 0.03), and the highest rate was in hematological (33.33%) and lung cancer (18.18%). Patients with metastatic cancer had a higher risk of death (OR 5.58, 95% CI 1.71–18.23, *p* = 0.01). The authors reported that patients with cancer were significantly more likely to require ICU admission (OR 2.84, 95% CI 1.59–5.08, *p* < 0.01) and have higher rates of severe/critical symptoms (OR 2.79, 95% CI 1.74–4.41, *p* < 0.01). A similar study of Deng et al. on 44,672 Chinese patients demonstrated that patients with cancer had a significantly higher risk of death than those without (RR 2.93, 95% CI 1.34–6.41, *p* = 0.006) [36]. Mehta et al. [37] analyzed data of 218 cancer patients with COVID-19; the mortality rate was 25% for patients with solid tumors and 37% for those with hematological malignancies. 

A systematic review and meta-analysis of Desai et al. [38] identified 11 studies of patients with COVID-19 and cancer; the authors found that 2% (95% CI, 2%–3%) of patients treated for COVID-19 had cancer. A similar systematic review and meta-analysis by Emami et al. identified 10 studies and found a prevalence of 0.92% of cancer in patients with COVID-19 (95% CI, 0.56%–1.34%) [39]. Another systematic review and meta-analysis of Wang et al. stated that cancer patients had an odds ratio of severe complications for SARS-CoV-2 infection of 2.29 (95% CI 1.00–5.23) [40]. A recent report of the WHO indicates 7.6% higher mortality for COVID-19 in cancer patients than in the general population [41]. 

In summary, some considerations could be made: first, particular attention should be paid to patients with active cancer and cancer survivors [41]; these patients have a high risk of infection due to lymphocytopenia, poor microcirculation, and an alteration of the intestinal bacterial flora. Second, recent anticancer therapies have an impact on the prognosis of COVID-19 because, after adjusting for risk factors (age, smoking history) and other comorbidities, cancer patients treated with chemotherapy or surgery within 2 weeks of the viral infection have a poor prognosis compared to patients not recently treated. A very interesting review of Zordoky B [42] described the cardiovascular vulnerability of cancer survivors with three scenarios for increased cardiovascular outcomes of COVID-19 in cancer patients. Firstly, cardiotoxic cancer treatment and COVID-19 synergize to exacerbate direct myocardial damages, mainly through pro-oxidative, pro-apoptotic and pro-inflammatory effects. Secondly, cardiotoxic cancer treatment leads to a reduced cardiac reserve in cancer survivors, making them more vulnerable to COVID-19. Finally, several shared risk factors may aggravate cardiovascular complications caused by both cancer treatment and COVID-19. Therefore, we highlight the cardiovascular outcomes in cancer patients with COVID-19, especially for those treated with cardiotoxic anticancer treatments that could exacerbate SARS-CoV-2-related cardiac effects. Particular attention should be paid to cancer patients treated with highly cardiotoxic therapies such as anthracyclines [43,44], human epidermal growth factor receptor 2 (HER2) blocking antibodies [45], tyrosine kinase inhibitors [46,47], proteasome inhibitors [48], as well as immune checkpoint inhibitors [49]. These drugs increased apoptosis, fibrosis, oxidative stress, and necrosis in myocardial cells, increasing the risk of heart failure and cardiomyopathies. Cardiovascular events increased in patients treated with combinatorial therapies; for example, HER2 and breast cancer patients treated with anthracyclines and trastuzumab [50] have a cumulative incidence of adverse cardiac events of 16.4, 23.8, and 28.2% after 1, 2, and 3 years of cancer diagnosis, respectively. A different picture should be seen in cancer patients treated with immune checkpoint inhibitors (ICIs) [51]. ICIs include blocking antibodies against programmed death 1 (PD-1) such as Nivolumab and Pembrolizumab; programmed death ligand 1 (PD-L1) with Atezolizumab, Avelumab, and Durvalumab or cytotoxic T-lymphocyte–associated antigen 4 (CTLA-4) inhibitors such as Tremelimumab and also Ipilimumab [52,53]

ICIs activate the immune response through the inhibition of CTLA-4, PD-1, or PDL-1-related pathways; outcomes in cancer patients are well documented with improved survival in patients with several cancers [52]. However, a broad spectrum of side effects has been recorded, involving vasculitis, venous thromboembolism, Takotsubo syndrome, atherosclerosis, myocarditis, and hypertension [54,55]. Actually, many discussions are open about the beneficial or harmful role of ICIs in patients with COVID-19 [51]. Initial considerations were made on the potential increase in fatal myocarditis in patients with COVID-19, as ICIs by themselves stimulate lymphocyte and macrophage infiltration into the myocardium [56,57]. Could the concomitant SARS-CoV-2 infection exacerbate this process? The authors conclude that multicenter retrospective studies will be required. However, more recent studies confirm that treatment with ICIs is not only safe in cancer patients with COVID-19, it could even be beneficial by exerting an immune-stimulating action [58].

## 4. Fatal Myocarditis 

Myocarditis is an uncommon, potentially life-threatening disease that presents with a wide range of symptoms in patients exposed to viral and bacterial agents and some drugs [59]. Considering that the major long-term consequence of myocarditis is dilated cardiomyopathy with chronic heart failure, patients with severe multiple organ dysfunction and fulminant myocarditis consequently had a high risk of mortality [59]. Known risk factors for viral myocarditis are high age, malnutrition (i.e., vitamin E and selenium deficiency as well as exposure to high doses of mercury), and sex hormones [59,60]. The pathogenesis of viral-induced myocarditis involves both innate and acquired immune response in myocardium: the innate immune response involves the release of nitric oxide and cytokines production (tumour necrosis factor, interferones, interleukins); the acquired immune response is based on the infiltration of monocytes, macrophages, B and T lymphocytes, antibodies, and autoantibodies [61] against cardiomyocytes [54,59]. Several studies associated COVID-19 to a high risk of fatal myocarditis, both in adults and children patients [8]. SARS-CoV-2 can cause inflammation of the heart tissue, i.e., myocarditis by recruiting cells of the immune system to cardiomyocytes [62]. Approximately 10% of hospitalized patients with COVID-19 have cardiovascular disease at the time of diagnosis, and myocarditis is often described [63]. Another recent study reported myocarditis associated to cardiogenic shock, heart failure, and valvulitis in three children with COVID-19 [64]. 

The pathogenesis of myocarditis and COVID-19 involves the same interleukins and cytokines [65]; in fact, the ICU COVID-19 patients described in the literature have high plasma cytokines that induced the recruitment of immune cells in the myocardium [59]. Fatal myocarditis is well known in oncology, considering that some anticancer drugs are well associated to innate or acquired immune response in myocardial tissues, as described in Section 3 [54,55]. Notably, myocarditis has a prevalence of 0.06% and 2.4% in cancer patients treated with ICIs, especially when combined [66]. Although rare, ICI-induced cardiotoxicity is well associated to cardiomyopathy, myocarditis, and pericarditis [67]. A recent World Health Organization report called VigiBase described that cancer patients treated with ICIs had a high risk of myocarditis compared to other anticancer therapies (fatality rate 46% in combinatorial therapies compared to monotherapies) [68]. As an example, 0.06% of patients treated with nivolumab had myocarditis compared to 0.27% in patients treated with Ipilimumab in combination [69]. Another example of fulminant myocarditis is described after one year of Ipilimumab [70]. In addition, nivolumab was recently associated to acute lymphocytic myocarditis in patients with lung cancer [71]. Therefore, it is speculated that cancer patients treated with immunotherapies could have a higher risk of myocarditis during COVID-19, which is in line with scenarios of Zordoky [42]; however, the risk of myocarditis should be compared to the possible beneficial effects of ICIs treatments due to their immune-stimulating effects [58]; therefore, more retrospective studies are needed to measure the index of fatal myocarditis in these patients and overall risk of mortality and prognosis.

## 5. Venous Thromboembolism 

COVID-19 infected patients have a high risk of venous thromboembolism (VTE) [1]. In a retrospective study comparing survivors and non-survivors of COVID-19, high levels of fibrin degradation products and D-dimer were seen in dead subjects indicating events of VTE [72]; notably, most of them showed small blood clots that developed throughout the bloodstream, blocking small blood vessels (disseminated intravascular coagulation) [72]. In another Chinese study, a high risk of hospital death has been associated with high levels of D-dimer (>1 g/L), confirming the high risk of thrombosis in patients with COVID-19. A more recent study indicates that 25% of patients hospitalized in the ICU for COVID-19 had VTE [73]. It is conceivable that anticancer therapies associated to VTE could exacerbate intravascular coagulative damages induced by SARS-CoV-2 infection. In fact, patients with active cancer have a higher risk of VTE and bleeding (both symptomatic or incidental), especially in those treated with radiotherapy and pro-inflammatory drugs [74,75]. A very impressive case report was recently published by Thompson et al. [76], with evidences of cerebral venous sinus thrombosis with hemorrhagic infarct associated with COVID-19 in a 50-year-old man; a computed tomography scan clearly showed left transverse sinus thrombosis, extensive superior sagittal sinus thrombosis, and hemorrhagic infarct and surrounding edema in the left temporal lobe in this patient. A recent retrospective cohort study of 147 American patients with COVID-19 reported that the overall incidence of VTE was 17%, the all-cause mortality in the VTE group was greater than that in the non-VTE group (48% vs. 22%; *p* = 0.007) [77]; also in this case, the authors concluded that patients with invasive mechanical ventilation and an admission D-dimer level ≥ 1500 ng/mL had a high risk of VTE. 

However, close attention must be given to cancer patients, considering that some anticancer therapies are associated to a high risk of VTE [78]. Patients with active cancer face higher risks of VTE, atrial arrhythmias, and bleeding events [79]. The risk of VTE and bleeding depends on the type of cancer and the followed therapies. Pancreas, stomach, and metastatic cancers have a very high risk of thrombotic adverse events; gynecological, lung, brain, hematological, and genitourinary (excluding prostate cancer) are considered as high risk for VTE; breast, prostate, and colon cancers are associated to a modest risk of VTE [79]. Many cancer therapies are associated to a high risk of VTE [79]. Surgery is associated with a 2-fold increased risk of postoperative VTE and a 4-fold increased risk of pulmonary embolism (PE)-related death in cancer. Select cancer therapies associated with thrombotic adverse events include platinum-based drugs (i.e., cisplatin), hormonal therapy (i.e., Tamoxifen), Anti-VEGF therapies (i.e., Bevacizumab, Sunitinib, Pazopanib), Bcr-Abl tyrosine-kinase inhibitors (TKI) (i.e., Nilotinib, Ponatanib), immunomodulators (i.e., Thalidomide, Lenalidomide and Pomalidomide), and proteasome inhibitors (i.e., Carfilzomib), as shown in Figure 4 [79]. Therefore, SARS-CoV-2 infection in these patients could exacerbate the thromboembolytic and bleeding processes already underway [80]. Therefore, a proper thromboprophylactic regimen for cancer patients with COVID-19 should be promoted, and the use of low molecular weight heparins (or unfractionated heparin) with or without mechanical prophylaxis should be preferred in acutely ill hospitalized patients.

## 6. Coagulation Dysfunctions: What about the Impact of Anticancer Therapies and Type of Cancer?

Several evidences indicated that COVID-19 increases the risk of coagulation dysfunctions [81]. Virus infection changed renin angiotensin system (RAS) and affected the fibrinolytic pathways [81]. As described in Figure 4, SARS-CoV-2 reduced the availability of angiotensin II, increased Plasminogen Activator Inhibitor-1 (PAI-1), tissue plasminogen activator (tPA), and urokinase expression, leading to increased platelet adhesion and coagulation. Moreover, increased antibodies against cardiolipin and β-2 glycoproteins in endothelial cells with high risk of lupus anticoagulant were also described. Moreover, SALR-CoV-2 activates Damage-associated molecular patterns (DAMP) in endothelial cells, leading to the recruitment and activation of monocytes (by increasing phosphatidylserine residues). Notably, the virus changed the extracellular matrix in the endothelium through the reduction of glcocalyx and antithrombin factors on endothelial cells, leading to increased coagulation. Therefore, based on these mechanisms, coagulation dysfunction is a candidate risk factor for adverse outcomes in COVID-19 and should be carefully addressed in clinical practice [82]. A recent retrospective observational study [83] evidenced in 429 patients with SARS-CoV-2 infection that the overall and major bleeding rates were 4.8% and 2.3%, respectively. In the critically ill, the major bleeding rate was 5.6% (95% CI, 2.4–10.7) and associated to high D-dimer levels [83]. Coagulative dysfunctions are frequently seen in cancer patients, especially in long survivors [84]; for example, a coagulation homeostasis may be impaired after preoperative radiotherapy, which produces lesions leading to VTE and bleeding [85]. Chemotherapies could affect liver metabolism and affect the production of pro-coagulative factors; in fact, endothelial lesions may persist for many years after the end of treatments [85]. Notably, the risk of VTE was high in cancer survivors compared to non-cancer patients and adjusted hazard ratios ranged from 1.7 in prostate cancer patients to 9.7 in patients with pancreatic adenocarcinoma [86]. Recent updated trials stated that coagulation dysfunctions are more frequent in younger colorectal and breast cancer patients than the others, and these effects persists at 5 years after cancer diagnosis [86]. Especially for breast cancer patients, coagulation dysfunctions are associated to chemotherapies, long-term endocrine therapies alone, or associated to targeted therapies [87]. Bleeding is frequently described in patients with advanced cancer (10% of patients experienced at least one episode of bleeding coming to a 30% in patients with blood tumors [88]. The most studied anticancer drugs associated to bleeding episodes are targeted therapies [89]; for example bevacizumab and axitinib are associated to the highest and lowest bleeding risk, respectively [89]. Therefore, it is evident that some anticancer treatments are able to alter the coagulation, leading to blood clotting and sometimes resulting in severe or fatal stroke; therefore, a relevant topic is the exacerbation of the coagulation dysfunctions in cancer patients with COVID-19 compared to non-cancer patients. Therefore, a close analysis of coagulation homeostasis in cancer patients at high risk of bleeding or VTE should always be promoted.

## 7. Angiotensin-Converting Enzyme 2 and Serine Protease TMPRSS2: Role in COVID-19 and Involvement in Cardio-Metabolic Risk Factors

Coronavirus binds to angiotensin-converting enzyme 2 (ACE2) and internalizes in target human cells through TMPRSS2 (a serine proteases useful for fusion of virus with host cells), replicating within them [90,91]. ACE2 and TMPRSS2 are key players of SARS-CoV-2 infection, and their expression depends on genetic and non-genetic factors [92,93]. Firstly, the coding gene for ACE2 is highly expressed in several organs, as demonstrated recently in a single-cell RNA sequencing study (scRNA-seq), i.e., the respiratory tract, lungs, esophagus, kidneys, bladder, the small intestine and heart; in fact, the clinical manifestation of the infection mainly involves these organs [92]. Other organs with a lower expression of ACE2 are nasal moss, the bronchial tree, liver, and stomach. This differential analysis of ACE2 expression could differentiate the clinical outcomes and severity of the patients’ symptoms, considering the correlation between ACE2 receptor density and COVID-19 uptake in human cells. Moreover, ACE2 gene polymorphisms could differentiate the risk of stroke, hypertension, or diabetes as well as viral-infection risk, depending on different ethnic groups [92]. 

In addition, patients treated with ACE inhibitors have a greater expression of ACE2 in heart and lungs, so these patients may have a higher risk of COVID-19 infection compared to others [93]. However, researchers recently proposed a possible therapeutic use of angiotensin type 1 antagonists such as losartan as a possible treatment for COVID-19 patients, considering the following mechanism [5]: during virus infection, after the spike protein binding to ACE2, for a feedback mechanisms phenomena, the human body increases the ACE production, thus stimulating also the ACE1 receptor (a well-known enhancer of pulmonary vascular permeability in humans), thereby promoting lung injury in these patients [94]; for this reason, the inhibition of ACE1 receptor is currently under discussion for a possible reduction of pulmonary vascular permeability in COVID-19 patients, thus improving clinical outcomes.

Tobacco use increases the gene expression of ACE2, which could explain the elevated susceptibility to COVID-19 in smokers [95]. Some preclinical research demonstrated that obesity increases ACE2 in adipose tissues and other organs [96]; this effect is in line with an association of high invasive mechanical ventilation for SARS-CoV-2 in obese (BMI > 30 kg/m^2^) and severely obese (BMI > 35 kg/m^2^) patients [97]. Other considerable factors involved in ACE2 expression could be the use of some anti-inflammatory drugs such as ibuprofen [98] and PPAR (peroxisome proliferator-activated receptors) activators, which are conventionally used for the treatment of diabetes mellitus type 2, including thiazolidinediones [98]; however, their role in risk infection was hypothesized, and more studies are needed to confirm this hypothesis. Another key regulator of ACE-2 expression is interleukin-17, which is able to maintain angiotensin II-induced hypertension and vascular dysfunction [99]; in association with interleukin 1-β (IL1-β), these cytokines induce myocarditis associated to viral infection and the use of immune checkpoint inhibitors [100,101].

Serine protease TMPRSS2 activates the severe acute respiratory syndrome coronavirus spike protein for membrane fusion [102]. TMPRSS2 expression on lungs, cancer cells, and cardiomyocytes is well studied [103]. Notably, TMPRSS2 expression depends on androgen levels [104]. Several cardiovascular risk factors such as obesity, metabolic syndrome, visceral fat, and the Western diet increases androgen levels in male and female subjects; therefore, we speculate that they could enhance TMPRSS2 expression in human cells. However, studies on the direct relationship between obesity and nutrition with TMPRSS2 expression have never been made. Interestingly, Montopoli et al. [34] described that prostate cancer patients receiving androgen-deprivation therapy (ADT) have a lower risk of SARS-Cov-2 infection than untreated ADT prostate cancer, indicating the key importance of androgens in the pathogenesis of COVID-19.

## 8. Cardiovascular Microenvironment: A Focus on Cytokine Storm 

SARS-CoV-2 infection stimulates the overproduction of pro-inflammatory cytokines, chemokines, and growth factors [1] involved in the pathogenesis of multi-organ damages, including severe pneumonia, myocarditis, heart failure, and VTE. As summarized in Table 2, COVID-19 patients have high plasma levels of a specific set of cytokines: Interleukin (IL) 1-β, IL-1RA, IL-7, IL-8, IL-9, IL-10, basic fibroblast growth factor (FGF), Granulocyte colony-stimulating factor (GCSF), Granulocyte-monocyte colony-stimulating factor (GMCSF), Interferon γ (INF), Interferon gamma-induced protein (IP10), monocyte chemoattractant protein 1 (MCP1), Macrophage Inflammatory Proteins (MIP) 1A, MIP1B, Platelet-derived growth factor (PDGF), Tumor necrosis factor (TNF)-α, and Vascular-Endothelial Growth Factor (VEGF) [1]. In addition, ICU patients have significantly higher IL-2, IL-7, IL-10, IL-12, GCSF, IP10, MCP1, MIP1-α, and TNF-α levels than non-ICU patients. The cytokine storm is the key player of COVID-19-associated outcomes in patients. To date, no data are available to differentiate plasma cytokines during COVID-19 in patients with cancer and patients without cancer. However, these effects lead to three considerations: first, SARS-CoV-2 changes the lung and cardiovascular microenvironment through a specific set of cytokines; second, cytokines are involved in myocardial injuries, especially in myocarditis, heart failure and arrhythmia, and VTE, which are the most seen causes of death in patients with COVID-19 [17]; third, cytokine-blocking agents could be the first-line regimen for pulmonary distress and cardiovascular diseases associated to COVID-19. A brief summary on the role of cytokines involved in the pathogenesis of COVID-19 and their role in cardiovascular diseases is proposed (Table 2):

Interleukin-1β, significantly increased in COVID-19 patients vs. healthy subjects, is one of the most important and studied cardiovascular risk factors involved in the NLRP3 inflammasome activation [105]; it has been studied for many years in cardio-oncology, in fact its expression is enhanced during doxorubicin-induced cardiotoxicity [106,107]. Some IL-1 blocking antibodies are currently under study and used in clinical trials with great improvements of cardiovascular outcomes [108]. Moreover, IL-1 promotes the expression of other pro-inflammatory cytokines, including interleukin-6 and cardiovascular risk factors such as hs-CRP [109]. 

Interleukin-2, overexpressed in both ICU-COVID-19 and non-ICU COVID-19 patients, indicating a Th1 immune reaction to SARS-CoV-2, is a T-cell growth factor of key importance in immune-reactive processes. High levels of IL-2 receptor are associated to rheumatoid arthritis [110], multiple sclerosis, and coronary artery diseases [111]; two case reports described myocarditis induced by an overexpression of myocardial IL-2 [112,113]. 

Interleukin-7 (IL-7) is a key regulator of T-cell growth [114]; IL-7 recruits monocytes and macrophages to the endothelium and plays a crucial role in the pathogenesis of atherosclerosis through the Phosphoinositide 3-kinases (PI3K)-AKT and nuclear factor kappa-light-chain-enhancer of activated B cells (NF-kB) pathways [114]; in another study, IL-7 promoted clinical instability in patients with coronary artery disease [115]. 

Interleukin 6, which is significantly increased in COVID-19 patients compared to healthy subjects (but without differences compared to ICU patients), is another key regulator of immune-related reaction. IL-6 is studied in oncology and cardioncology because it is a key promoter of cancer survival, chemoresistance, anticancer-induced cardiotoxicity [106], and cancer progression [116]. Meta-analysis confirmed the association between IL-6 receptor and coronary heart diseases [117], and in a recent study in 2329 patients with heart failure, high plasma levels of IL-6 (seen in 50% of patients) were associated to atrial fibrillation, a reduction of Left Ventricular Ejection Fraction (LVEF), and a worse prognosis [118].

Interleukin-12 acts as protective cytokine in anticancer-induced myocardial injuries, although some authors discussed its possible association to heart failure or atherosclerosis [119]; however, the role of IL-12 remains controversial and needs further observational and interventional studies. 

Granulocyte colony-stimulating-factor (G-CSF) is produced by leukocytes and fibroblasts; it is well associated to a higher risk of MACE (death, myocardial infarction, re-hospitalization) in patients with stable coronary artery disease [120], although other authors discussed its cardioprotective role through the induction of tissue repair after myocardial infarction [121]. 

C-X-C motif chemokine 10 (CXCL10 or induced protein 10) is a chemoattractant chemokine of Th1 and cytotoxic T cells. It is overexpressed in viral myocarditis [122], coronary atherosclerosis [123], hypertension [124], and left ventricular dysfunction [125]. 

CCL2 (Monocyte Chemoattractant Protein, called also MCP-1) is another well-known cytokine in cardiology that is strictly related to several cardiovascular events such as atherosclerosis, myocardial injury, hypertension, angiotensin-2 homeostasis (through functional interaction with an angiotensin2 type 1 receptor) [126], and other diseases. 

TNF-α, which has been significantly associated to severe cases of COVID-19, has been studied as a driver of vascular dysfunction, atherosclerosis, and heart failure [127]. 

As stated before, 90% of patients with COVID-19 [128] had high d-dimer, a fibrin degradation product, which is defined as an independent risk factor for cardiovascular mortality [129]; all these cytokines act in concert in the initiation and progression of endothelial damage, stimulating the release of pro-coagulant factors associated to ischemia and thrombosis. Therefore, anti-cytokine therapies should be studied and promoted in order to improve the hemodynamic profile in patients with COVID-19.

In summary, it is speculated that cytokine-related pathways will be potential targets in multiorgan damages during COVID-19. The NLRP3 (NOD-, LRR- and pyrin domain-containing protein 3) and MyD88 (myddosome) are key mediators upstream of cytokines storm [130] involved in acute respiratory distress syndrome, VTE, myocarditis, and heart failure in COVID-19. NLRP3 and MyD88 induce the release of IL-1 and hs-CRP [131], and several studies associated NLRP3 activation in multiorgan damages of virus infection [132]. Therefore, oral NLRP3 inhibitors [133,134,135] clinically used for therapy of severe gout, arthritis, and prevention of heart failure (i.e., dapansutrile) are currently proposed in the USA as a strategy to prevent acute respiratory distress syndrome and cardiovascular diseases in COVID-19 [136]. 

Another cytokine-blocking agent currently under study in COVID-19 is the antibody against IL-1 (canakimumab), which is well known by cardiologists as a cardioprotective agent in high-risk patients [109]; another drug under study is tocilizumab, which is an antibody that competitively inhibits the binding of IL-6 to its receptor (IL-6R) just used by oncologists to reduce pulmonary distress in cancer patients during immune checkpoint inhibitors [137,138]. Another strategy is the administration of interleukin-37 (IL-37), a member of IL-1 family [139], which is able to reduce the MyD88/NLRP3-IL-1/IL-6 pathway [140], ameliorating pneumonia [141], viral myocarditis [142] (through Th17/regulatory T cell immune response), and other infectious diseases. Among the therapeutic strategies available, 453 high-risk patients with COVID-19 are currently treated with Remdesivir (GS-5734), which is a prodrug of adenosine nucleotide analog that interferes with the action of viral RNA polymerase, in a phase 3 randomized, double-blind, placebo-controlled trial in China [143]; as an antiviral agent, a proper cardiovascular control and management of these patients should be also considered, in light of the possible cardiotoxic events seen during some antiviral treatments, such as arrhythmia and cardiac dysfunction [144].

## 9. Non-Pharmacological Strategies to Reduce Cardiovascular Events in Cancer Patients with COVID-19

A proper management of preventive measures should be promoted in cancer patients, considering their systemic immunosuppressive state and cardiovascular vulnerability [145,146,147].

(1) First of all, cancer patients should be stimulated to perform daily physical activity, compatibly with their state of health, following WHO recommendations [148]. Cancer survivors doing daily physical exercises reduced cancer mortality by 27% compared with sedentary patients [149]. An appropriate physical activity reduced cardio-metabolic comorbidities in cancer patients and inflammation markers involved in heart failure and VTE [150]. 

(2) Patients with cancer should follow World Cancer Research Fund/American Institute for Cancer Research (WCRF/AICR) recommendations [150]. Briefly, WCRF/AICR principles are:-Reduce total body fat, especially visceral fat; pay attention to body weight;-Do daily exercise, at any time of the day; reduce the time spent on television;-Limit consumption of energy-dense foods and avoid sugary drinks;-Eat mostly foods of plant origin and follow a diet rich in whole grains, vegetables (non-starchy), fruit, and legumes;-Limit the consumption of red meats (beef, pork, sheep), cured meats, and preserved meats;-Limit alcohol consumption;-For cancer prevention, do not use supplements. Try to meet nutritional needs through diet alone;-For mothers: if you have the opportunity to breastfeed, this has benefits for the baby and the mother.

A recent meta-analysis of observational studies demonstrated that the adherence to WCRF/AICR recommendations reduced risks of cancer incidence and mortality in breast, lung, upper aerodigestive tract, stomach, prostate, and colorectal cancers [151,152,153], and they also improved hemostatic factors implicated in chronic disease development [154]. A special note on fiber intake and mortality should be made: there is an inverse association between fiber intake from foods and risk of death for respiratory diseases, infection, and cardiovascular diseases [155]. A recent report on 219,123 men and 168,999 women in the USA demonstrated that dietary fiber intake was associated with a lower risk of total death (multivariate RR comparing the highest vs. the lowest quintile = 0.78, 95% CI: 0.73–0.82, *p*-trend, <0.001 in men; 0.78. 95% CI: 0.73–0.85, *p*-trend, <0.001 in women). Dietary fiber intake (derived from whole grains, as suggested in WCRF/AICR recommendations) also lowered the risk of death from cardiovascular disease and respiratory diseases by 24%–56% in men and 34%–59% in women [156]. However, to date, no studies associated fiber intake with risk of mortality rate in patients with COVID-19.

(3) Cancer patients should reduce contact with external people as much as possible, promoting self-quarantine status, telemedicine, and minimizing medical examinations after a proper consult with your oncologist and cardiologist. To reduce the rate of transmission among patients with stable cancer and cardiovascular diseases, substituting in-person visits with telehealth visits and deferring any non-urgent procedures should be strongly considered. 

(4) In cancer patients treated with cardiotoxic anticancer drugs, following the American Heart Association Scientific Statement on Cardio-Oncology Rehabilitation [157] to reduce cardiovascular outcomes in cancer patients and survivors should be strongly promoted. In brief, this statement from the American Heart Association provides an overview of the existing knowledge and rationale for the use of cardiac rehabilitation to provide structured exercise and ancillary services to cancer patients and survivors in order to reduce risk of mortality.

(5) In cancer patients with high cardiovascular risk, a proper hemodynamic analysis and control of biomarkers of heart damage associated to echocardiographic studies should be promoted as a primary strategy of cardiovascular prevention during COVID-19. 

(6) Control of glycemic homeostasis in these patients should be promoted in hospital practice involved in the management of COVID-19, considering that almost all the observational studies available correlate diabetes or hyperglycemia with a poor prognosis of COVID-19 [158].

## 10. Conclusions

Currently available data indicate that metabolic syndrome, obesity, previous cardiovascular diseases, diabetes mellitus, or hyperglycemia are prognostic risk factors in patients with COVID-19. Patients with cancer and cardiovascular diseases are particularly vulnerable and classifiable as high-risk patients. As a community, we should pay close attention to these patients, especially to those recently treated with cardiotoxic anticancer therapies or radiotherapy associated to heart failure, myocarditis, and VTE. A close cardiovascular and hematological screening should be promoted in high-risk patients, stimulating them to follow WCRF/AICR recommendations based on a healthy and physically active lifestyle.

## Figures and Tables

**Figure 1 cancers-12-03316-f001:**
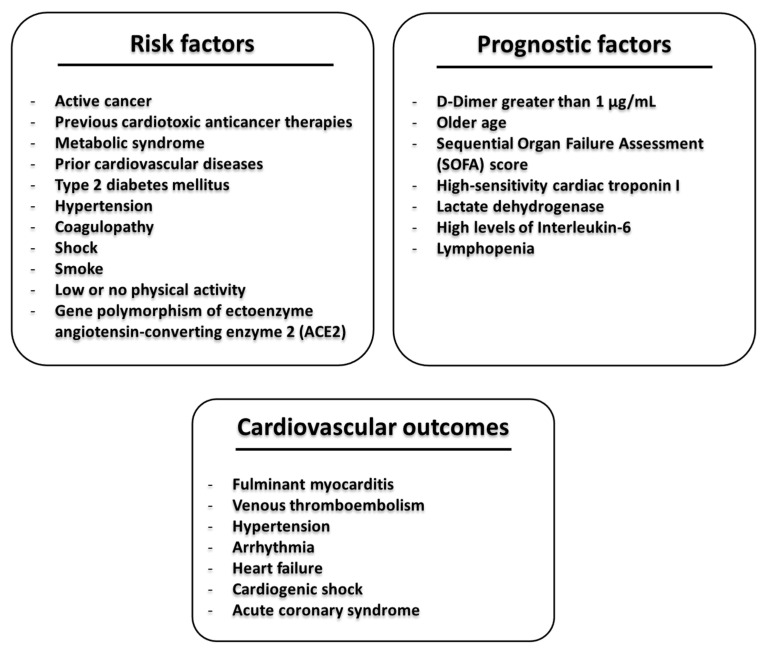
Cardiovascular adverse outcomes in patients with cancer during Severe Acute Respiratory Syndrome coronavirus 2 (SARS-CoV-2) infection are exacerbated by four interrelated risk factors: cancer itself, which causes immunosuppressive effects; cardiotoxic, pro-fibrotic, and pro-inflammatory anticancer therapies; history of cardiovascular diseases (hypertension, coronary heart disease, or diabetes); direct cardiac effects of SARS-CoV-2 infection.

**Figure 2 cancers-12-03316-f002:**
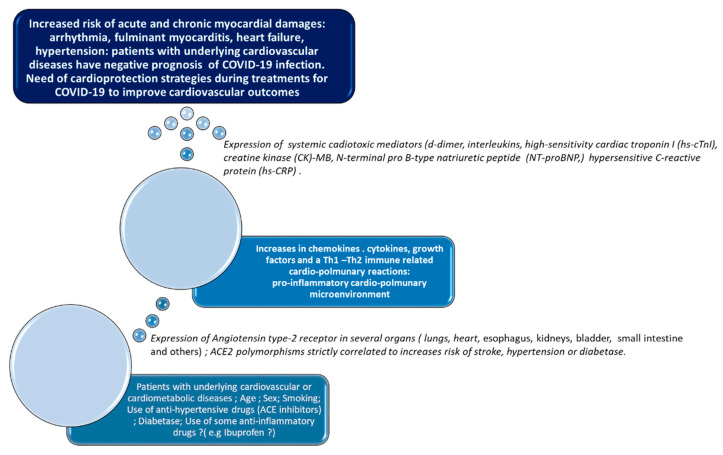
A representative scheme of the general cardio-metabolic risk factors, prognostic factors involved in coronavirus disease-2019 (COVID-19), and cardiovascular outcomes.

**Figure 3 cancers-12-03316-f003:**
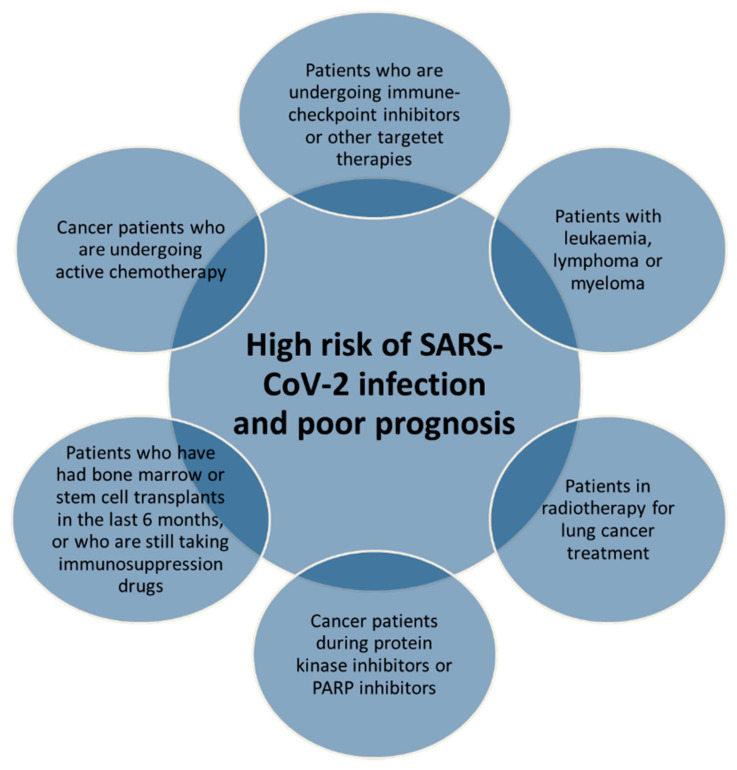
Overall risk factors for cancer patients according to “Clinical guide for the management of noncoronavirus patients of National Health Service England” [20].

**Figure 4 cancers-12-03316-f004:**
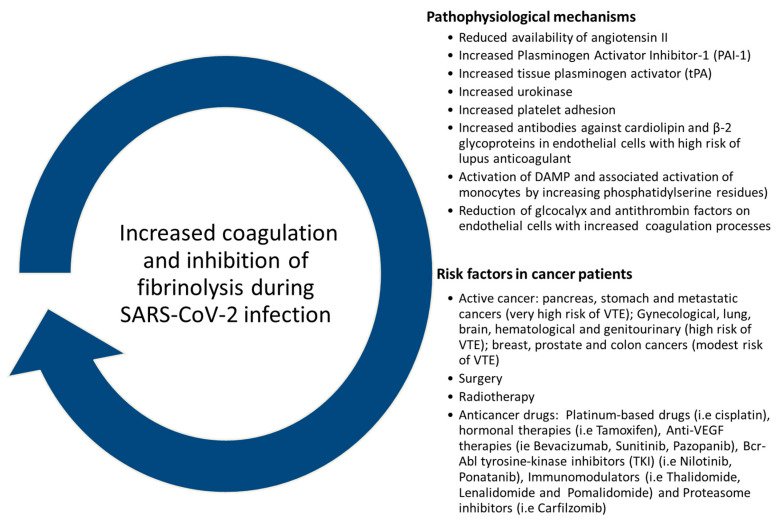
Pathophysiological mechanisms related to the coagulation dysfunctions in patients with COVID-19 and established risk factors in cancer patients that could increase coagulation and reduce the fibrinolysis.

**Table 1 cancers-12-03316-t001:** Retrospective clinical studies on cancer patients affected by SARS-CoV-2 infection.

Study	Date of Publication	Total Patients (*n*)	Type of Population	Outcomes in Cancer Patients	Reference
Liang et al.	14 February 2020	1580	Chinese	1% (95% CI 0.61–1.65) of COVID-19 cases had a history of cancer. Cancer patients have a high probability of severe events than non-cancer patients (hazard ratio 3.56, 95% CI 1.65–7.69). Risk of clinically severe events in cancer patients who underwent chemotherapy or surgery in the past month is higher than other cancer patients (odds ratio 5.34, 95% CI 1.80–16.18; *p* = 0.0026).	[21]
Jing Yu et al.	25 March 2020	1524	Chinese	Patients with cancer had a higher risk of SARS-CoV-2 infection (OR, 2.31; 95% CI, 1.89–3.02) compared with other people.	[22]
Guan et al.	26 March 2020	1590	Chinese	Cancer patients had a hazard ratio of 3.50 (95% CI, 1.60–7.64) for admission to intensive care unit, or invasive ventilation, or death, compared to non-cancer patients.	[23]
Istituto Superiore di Sanità	2 April 2020	909	Italian	Among 909 patients who died, more than 50% had three or more comorbidities such as ischemic heart disease (27.4%), atrial fibrillation (23%), heart failure (16.4%), stroke (12%), hypertension (73.5%), diabetes mellitus (31.5%), and active cancer in the past 5 years (16.5%)	[24]
Trapani et al.	28 April 2020	9	Italian	All cancer patients with COVID-19 had a smoking history, being either former smokers (*n* = 4) or current smokers (*n* = 5). Comorbidities were arterial hypertension (*n* = 4), diabetes mellitus type II (*n* = 2), and chronic kidney diseases.	[25]
Montopoli et al.	29 April 2020	9280	Italian	Cancer patients have an increased risk of SARS-CoV-2 infections than non-cancer patients. 8.5% had a diagnosis of cancer and 1.3% had prostate cancer. Comparing the total number of SARS-CoV-2 positive cases, patients with prostate cancer receiving androgen-deprivation therapy had a significantly lower risk of SARS-CoV-2 infections compared to patients who did not receive androgen-deprivation therapy (OR 4.05; 95% CI 1.55–10.59).	[26]
de Rojas et al.	8 May 2020	15	Spanish	The prevalence of COVID-19 infection among children with cancer in Madrid is 1.3% vs. 0.8% of the general pediatric population.	[28]
Wu et al.	24 February 2020	72314	Chinese	Case fatality report was higher among patients with pre-existing comorbidities: 10.5% for cardiovascular disease, 7.3% for diabetes, 6.3% for chronic respiratory disease, 6.0% for hypertension, and 5.6% for cancer.	[29]
Onder et al.	23 March 2020	1625	Italian	In a subsample of 355 COVID-19 patients who died in Italy, 30% had ischemic heart disease, 35.5% had diabetes, 20.3% had active cancer, 24.5% had atrial fibrillation, 6.8% had dementia, and 9.6% had a history of stroke.	[30]
Zhang et al.	26 March 2020	28	Chinese	The mortality rate of cancer patients was 28.6% (more than ten times higher than that reported in all COVID-19 patients in China). The recent use of anticancer therapies within 14 days of infection (including chemotherapy, immunotherapy, and radiation) was an independent predictor of death or other severe events with a hazard ratio > 4.	[31]
Bonomi et al.	31 March 2020	Single report	Italian	A 65-year-old patient with metastatic non-small-cell lung cancer in treatment from 6 years with anti PD-1 antibody (nivolumab). After COVID-19 diagnosis and hospitalization, the patient presented a rapid evolution of respiratory failure and was not treated with more invasive procedures, probably due to his cancer and emphysema history. The authors discuss the need for a multidisciplinary approach for the management of COVID-19-infected cancer patients.	[32]
He et al.	1 April 2020	128	Chinese	There was no significant difference in the proportion of patients with hematological cancers vs. healthy subjects (10% vs. 7%, *p* = 0.322), but a significant difference was seen in fatality rate (62% vs. 0%, *p* = 0.002). Among cancer patients, 39% had acute myeloid leukemia, 20% had acute lymphoblastic leukemia, 59% had received chemotherapy in the last 7–19 days.	[33]
Miyashita et al.	21 April 2020	334	American	Patients with cancer were significantly more likely to require intubation (RR 1.89, 95% CI 1.37–2.61) than non-cancer patients. Cancer patients younger than 50 years of age had a significantly higher rate of mortality than the other (RR 5.01, 95% CI 1.55–16.2).	[34]
Dai et al.	28 April 2020	641	Chinese	Patients with cancer had a higher mortality rate than those without (odds ratio 2.34, 95% CI 1.15–4.77, *p* = 0.03. The death rate was highest in those with hematological cancer (33.33%) and lung cancer (18.18%). Patients with metastatic cancer had a higher risk of death (OR 5.58, 95% CI 1.71–18.23, *p* = 0.01). Patients with cancer were significantly more likely to require ICU admission (OR 2.84, 95% CI 1.59–5.08, *p* < 0.01) and have higher rates of severe/critical symptoms (OR 2.79, 95% CI 1.74–4.41, *p* < 0.01).	[35]
Deng et al.	28 April 2020	44672	Chinese	Patients with cancer had a significantly higher risk of death than those without (RR 2.93, 95% CI 1.34–6.41, *p* = 0.006)	[36]
Mehta et al.	1 May 2020	218	American	The mortality rate in cancer patients with COVID-19 was 25% for solid tumors and 37% for hematological malignancies.	[37]

**Table 2 cancers-12-03316-t002:** Cytokines and chemokines storm induced by COVID-19 and differences between healthy subjects, Intensive Care Unit (ICU), and non-ICU, COVID-19 patients. Data and *p* values from C. Huang et al., Lancet 2020 [1].

Cytokine	Involvement in Pathogenesis of COVID-19 and Cardiovascular Diseases	References
Interleukin-1β (IL-1β)	Interleukin-1β is significantly increased in COVID-19 patients vs. healthy subjects, and it is one of the most important and studied cardiovascular risk factor involved in the NLRP3 inflammasome activation; it has been studied for many years in cardio-oncology; in fact, its expression is enhanced during doxorubicin-induced cardiotoxicity. Some IL-1-blocking antibodies are currently under study and used in clinical trials with great improvements of cardiovascular outcomes. Moreover, IL-1 promotes the expression of other pro-inflammatory cytokines, including interleukin-6 and cardiovascular risk factors such as hs-CRP.	[105,106,107,108,109]
Interleukin-2 (IL-2)	Interleukin-2, overexpressed in both ICU-COVID-19 and non-ICU COVID-19 patients, indicating a Th1 immune reaction to SARS-CoV-2, is a T-cell growth factor of key importance in immune-reactive processes. High levels of IL-2 receptor are associated to rheumatoid arthritis multiple sclerosis and coronary artery diseases; two case reports described myocarditis induced by an overexpression of myocardial IL-2.	[110,111,112,113]
Interleukin-7 (IL-7)	Interleukin-7 (IL-7) is a key regulator of T-cell growth; IL-7 recruits monocytes and macrophages to the endothelium and plays a crucial role in the pathogenesis of atherosclerosis through PI3K-AKT and NF-kB pathways; in another study, IL-7 promoted clinical instability in patients with coronary artery disease.	[114,115]
Interleukin 6 (IL-6)	Interleukin 6, which is significantly increased in COVID-19 patients compared to healthy subjects (but without differences compared to ICU-patients), is another key regulator of immune-related reaction. IL-6 is studied in oncology and cardio-oncology because it is a key promoter of cancer survival, chemoresistance, anticancer-induced cardiotoxicity, and cancer progression. Meta-analysis confirmed the association between IL-6 receptor and coronary heart diseases, and in a recent study in 2329 patients with heart failure, high plasma levels of IL-6 (seen in 50% of patients) were associated to atrial fibrillation, a reduction of Left Ventricular Ejection Fraction (LVEF), and a worse prognosis.	[116,117,118]
Interleukin-12 (IL-12)	Interleukin-12 acts as a protective cytokine in anticancer-induced myocardial injuries, although some authors discussed on its possible association to heart failure or atherosclerosis; however, the role of IL-12 remains controversial and needs further observational and interventional studies.	[119]
Granulocyte-colony-stimulating-factor (G-CSF)	Granulocyte colony-stimulating-factor (G-CSF) is produced by leukocytes and fibroblasts; it is well associated to a higher risk of MACE (death, myocardial infarction, re-hospitalization) in patients with stable coronary artery disease, although other authors discussed its cardioprotective role through the induction of tissue repair after myocardial infarction.	[120,121]
C-X-C motif chemokine 10 (CXCL10 or Induced protein 10)	C-X-C motif chemokine 10 (CXCL10 or induced protein 10) is a chemoattractant chemokine of Th1 and cytotoxic T cells. It is overexpressed in viral myocarditis, coronary atherosclerosis, hypertension, and left ventricular dysfunction.	[122,123,124,125]
CCL2 (Monocyte Chemoattractant Protein (MCP-1)	CCL2 (Monocyte Chemoattractant Protein, called also MCP-1) in another well-known cytokine in cardiology, strictly related to several cardiovascular events such as atherosclerosis, myocardial injury, hypertension, angiotensin-2 homeostasis (through functional interaction with angiotensin2 type 1 receptor), and other diseases.	[126]
Tumour Necrosis Factor-α (TNF-α)	TNF-α, significantly associated to severe cases of COVID-19, has been studied as a driver of vascular dysfunction, atherosclerosis, and heart failure.	[127]

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
