# Peer review of "SARS-CoV-2 Infection and Cardioncology: From Cardiometabolic Risk Factors to Outcomes in Cancer Patients"

_cancers, 2020, doi:10.3390/cancers12113316_

Round 1

Reviewer 1 Report

The authors presented an up-to-date review article on the cardiovascular risk of cancer patients after COVID-19 infection. In particular, the authors well described the cardiovascular risk of cancer patients focusing the attention on the expression of SARS-CoV-2 cellular receptors (ACE2 and TMPRSS2) and the incidence of cytokine storm, however, some parts should be better described and information about the alteration of the coagulative cascade observed in COVID-19 and cancer patients should be added. Below are reported some minor comments that will improve the quality of the manuscript:

1) In section “3. Outcomes in cancer patients”, please provide more details about the risk of severe COVID-19 symptoms in cancer patients undergoing immunotherapy. Indeed, recent studies and reviews indicate that cancer patients treated with ICIs have not an increased risk of severe symptoms. Please better argue this aspect. For this purpose, see:

– 10.3390/cancers12082237

– 10.1136/jitc-2020-001145

- 10.1136/jitc-2020-000933

2) Section 4 and 5 should be improved. Please see the following recent findings that could improve this section:

- 10.1016/j.fct.2020.111769

- 10.1097/CRD.0000000000000341

- 10.1136/practneurol-2020-002678

- 10.1016/j.jvsv.2020.09.006

3) The authors should add a new section on the alteration of the coagulative cascade often observed in COVID-19 patients. Some anticancer treatments are able to alter the coagulation leading to blood clotting sometimes resulting in severe or fatal stroke. In this context, it was observed that COVID-19 patients have abnormal levels of d-dimer, prothrombin and other coagulative markers. Please add relevant information about this topic. For this purpose, see:

- 10.1182/blood.2020006520

- 10.1016/j.blre.2020.100745

- 10.1182/blood.2020006000

4) The manuscript needs English editing performed by an English native speaker.

Author Response

We thank the reviewer for the proper comments and suggestions that are useful to us for our knowledge and to improve the quality of our work. We are totally agree with you and thank you for the suggestions on the vascular effects of SARS-CoV-2 infection. We think that these information will enhance the quality of the manuscript, updating it. Specifically,

  • Ok, we added these information in the revised manuscript file In section “3. Outcomes in cancer patients”. You are right, ICI could have different effects (harmful or beneficial ) in cancer patients with COVID-19, we highlight on this issue with the immune-stimulating effects of ICIs against virus ( pag 7-8, line 268-283) as well as the possible harmful effects if these treatments ( possible increase in the incidence of myocarditis? We described these potential effects in pag 11, line 291-324) and inserted new references as suggested by you. (ref 51-58 and 59-42)
  • Ok, we improved section 4 and 5 as you suggested. Specifically, we added more data on fatal myocarditis with mechanisms and the involvement of anticancer therapies ( ICIs) on its incidence ( with particular attention on nivolumab and ipilimumab and combined therapies) (pag 11 line 295-324 and new references 59-42); moreover, in section 5, we added more information about VTE risk and COVID-19, cancer (type of cancer) and the involvement of anticancer therapies ( radiotherapy, targeted therapies, hormonal therapies…). ( pag 11,12 , line 336-362; more references on this field are 73-80)
  • Thank you for this suggestion, as you requested, we added a new section titled “Coagulation dysfunctions: what about the impact of anticancer therapies and type of cancer? “ at pag 12,13 ( line 365-406) based on the effects of COVID-19 on coagulation dysfunctions, the underlying mechanisms ( the role of angiotensin II, PAI, tPA, D-dimer and glycocalyx ) and the involvement of anticancer therapies on the risk of bleeding ( also summarized in a new Figure, figure 4, pag 13; references on this field are from 81 to 89.
  • Yes, we revised the manuscript to improve the English

Reviewer 2 Report

It is a very interesting work, very well documented and very well written. The authors do an extensive review and present the data clearly and concisely. Tables and figures are very appropriate and provide information that is not redundant with the text. The discussion and conclusions are appropriate to the evidence reviewed and the bibliography is up to date

Author Response

We thank the reviewer for the nice comments, we are happy that the manuscript is considered interesting and useful for the research community.

Reviewer 3 Report

Dear Authors,   Please find the comments for the manuscript cancers-969620 entitled "SARS-CoV-2 infection and cardioncology: from cardiometabolic risk factors to outcomes in cancer patients".     The main topics in this review paper written by Vincenzo Quagliariello et al. focused on cardiovascular side effects of COVID-19 and cardio-metabolic risk factors in non-cancer and cancer patients. The manuscript provided cardiovascular outcomes in non-cancer patients with COVID-19, summarized retrospective studies on cancer patients with COVID-19 and touched some cardio-metabolic factors in cardiac events/cardiovascular diseases. The paper provided decent information and some insights on (1) cancers and COVID-19 and (2) cardio-metabolic risk factors and cancers with appropriate references. A review written from perspective of cardio-metabolic risk factors in cancer patients with COVID-19 is very interesting and could draw attention from the readers. However, this review article provided little information and insights that connect cardio-metabolic risk factors and COVID-19, especially in the cancer patients with COVID-19 or SERS-CoV-2. Since many reviews describing (1) cancers and COVID-19 and (2) cancers and cardio-metabolic factors in cardiac events/cardiovascular diseases are available, I am not convinced that the advance in knowledge is sufficient to warrant publication in this journal, Cancers. In conclusion, the reviewer will not recommend accepting the current version of this article. This study could be better suited for publication in a more specialized journal, where I believe your work will have a much higher impact on the relevant research community.

Author Response

We thank the reviewer for the comments; we also thank the reviewer for appreciating the commitment and quality of the work. We are agree with you that there were papers available in literature describing the correlation between cancer or cardiovascular diseases and COVID-19. However, we would like to underline some innovative aspects of our manuscript that may be the point of attention in this journal:

1) we described the role of TMPRSS2 in viral pathogenesis and the possible correlation with cardiometabolic risk factors carrying out discussions on the role of anticancer therapies on their expression and this topic is not well described in literature.

2) We critically analyze the possible influences of anticancer therapies ( and the type of cancer) on the incidence and prognosis of myocarditis, VTE and bleeding ( in the revised manuscript file, we added more information on this field, through new data and updated references ( see pag 11 line 295-324 and new references 59-42 ; pag 11,12 , line 336-362; more references on this field are 73-80.

3) Non-pharmacological strategies to reduce cardiovascular events in cancer patients with COVID-19  ( pag 19, section 9) is partially discussed in literature, and it should get more attention from the scientific community ; we have made some non-pharmacological management proposals following the World Cancer Research Fund/American Institute for Cancer Research (WCRF/AICR) recommendations  (a topic that needs greater clarity because it is still poorly highlighted in scientific literature).

However, aimed to enhance the quality of the manuscript, updating it, we added more new information and data on other little-dealt aspects on the topic “COVID-19 and cancer” ( the changes were highlight using the "Track Changes" function in Microsoft Word); specifically:

  • in the revised manuscript file In section “3. Outcomes in cancer patients” we discussed on different effects (harmful or beneficial ) of ICIs in cancer patients with COVID-19; we highlight on this issue  with the immune-stimulating effects of ICIs against virus ( pag 7-8, line 268-283) as well as the possible harmful effects if these treatments ( possible increase in the incidence of myocarditis? We described these potential effects in pag 11, line 291-324) and added new references. (ref 51-58 and 59-42)
  • we improved section 4 and 5; specifically, we added more data on fatal myocarditis with mechanisms and the involvement of anticancer therapies (ICIs) on its incidence ( with particular attention on nivolumab and ipilimumab and combined therapies) (pag 11 line 295-324 and new references 59-42); moreover, in section 5, we added more information about VTE risk and COVID-19, cancer (type of cancer) and the involvement of anticancer therapies ( radiotherapy, targeted therapies, hormonal therapies…). ( pag 11,12 , line 336-362; more references on this field are 73-80)
  • Moreover, considering the important link between SARS-CoV-2 infection and coagulopathies and that many anticancer therapies (as well as some types of tumors) are associated with a high risk of VTE and bleeding, we added a new section titled “Coagulation dysfunctions: what about the impact of anticancer therapies and type of cancer? “ at pag 12,13 ( line 365-406) based on the effects of COVID-19 on coagulation dysfunctions, the underlying mechanisms ( the role of angiotensin II, PAI, tPA, D-dimer and glycocalyx ) and the involvement of anticancer therapies on the risk of bleeding ( also summarized in a new Figure, figure 4, pag 13; references on this field are from 81 to 89.

Based on these new information and studies, we hope that the current version of the manuscript is improved, updating it on topics still not clearly described in literature, making it publishable in this journal.

Round 2

Reviewer 3 Report

Dear Authors,

Please find the comments for the revised manuscript cancers-969620 entitled "SARS-CoV-2 infection and cardioncology: from cardiometabolic risk factors to outcomes in cancer patients".  

The revised manuscript written by Vincenzo Quagliariello et al. covered a good amount of previous cases and summarized cardiovascular side effects of COVID-19 and cardio-metabolic risk factors in non-cancer and cancer patient well. The sections added in the revised manuscript provided the possible influences of anticancer therapies and cancers on the incidence and prognosis of myocarditis, and it would give additional values through new insights in the field of myocarditis. I appreciated the points described in the authors comments and agree to some innovative aspects of the manuscript (points 1-3) and will recommend the publication of this manuscript with minor modification. Minor points to consider in subsequent versions are provided to the comments for the authors.

  1. Table 2 is not cited in the main text.
  2. Section 3 “Outcomes in cancer patients” provided retrospective studies on cancer patients (from line 158). The many reports introduced here handled different outcomes and factors. I understand that some reports handle multiple factors, but reorder of the information could give less confusion to the readers Ref 24 described the rate of infection, Ref 25 showed admission to ICU, Ref 26 provided mortality and types of cancers, Ref 27 gave a case report of non-small-cell lung cancer, Ref 28 described fatality rate and types of cancer. Ref 29 gave cardiovascular diseases, Ref 30 touched mortality, Ref 31 focused on some comorbidities, Ref 32 provides mortality and ICU admission, Ref 33 showed the risk of death and Ref 34 showed infection rates, Ref 36 gave mortality rates……  It could be challenging, but authors could consider rearranging order of the references based on outcomes or other criteria.

Author Response

Dear reviewer, thank you for the comments and suggestions. We are pleased that the current version of the manuscript has provided more information and details on the topic COVID-19, cancer and cardiovascular diseases. Thank you for the suggestions and the proper comments useful to improve the quality of the manuscript. We have made the changes as required, highlighted in green:

  • Ok, sorry for the mistake. we added the Table 2 in the manuscript (pag 4 and 5);
  • Ok, we are agree with you. in the previous version we listed the studies in chronological order, however, we are agree with you that listing the studies by topic makes the manuscript better readable. Based on this, we first included the studies related to the correlation COVID-19 / infection rate and comorbidity in cancer patients ( Table 1, pag 9, ref 21-28); studies that correlated COVID-19 with the mortality rate in cancer patients were subsequently listed ( Table 1, pag 9, ref 29-37). We updated references with these new sequences.We hope that the current version of the manuscript can be acceptable for publication in this journal.